# Development of a Predictive Model for Carbon Dioxide Corrosion Rate and Severity Based on Machine Learning Algorithms

**DOI:** 10.3390/ma17164046

**Published:** 2024-08-14

**Authors:** Zhenzhen Dong, Min Zhang, Weirong Li, Fenggang Wen, Guoqing Dong, Lu Zou, Yongqiang Zhang

**Affiliations:** 1College of Petroleum Engineering, Xi’an Shiyou University, Xi’an 710065, China; 2Shaanxi Key Laboratory of Carbon Dioxide Sequestration and Enhanced Oil Recovery, Xi’an 710075, China

**Keywords:** CO_2_ corrosion, machine learning, Pearson correlation coefficient, RF, XGBoost

## Abstract

Carbon dioxide corrosion is a pervasive issue in pipelines and the petroleum industry, posing substantial risks to equipment safety and longevity. Accurate prediction of corrosion rates and severity is essential for effective material selection and equipment maintenance. This paper begins by addressing the limitations of traditional corrosion prediction methods and explores the application of machine learning algorithms in ${CO}_{2}$ corrosion prediction. Conventional models often fail to capture the complex interactions among multiple factors, resulting in suboptimal prediction accuracy, limited adaptability, and poor generalization. To overcome these limitations, this study systematically organized and analyzed the data, performed a correlation analysis of the data features, and examined the factors influencing corrosion. Subsequently, prediction models were developed using six algorithms: Random Forest (RF), K-Nearest Neighbors (KNN), Gradient Boosting Decision Tree (GBDT), Support Vector Machine (SVM), XGBoost, and LightGBM. The results revealed that SVM exhibited the lowest performance on both training and test sets, while RF achieved the best results with R^2^ values of 0.92 for the training set and 0.88 for the test set. In the classification of corrosion severity, RF, LightGBM, SVM, and KNN were utilized, with RF demonstrating superior performance, achieving an accuracy of 99% and an F1-score of 0.99. This study highlights that machine learning algorithms, particularly Random Forest, offer substantial potential for predicting and classifying ${CO}_{2}$ corrosion. These algorithms provide innovative approaches and valuable insights for practical applications, enhancing predictive accuracy and operational efficiency in corrosion management.

## 1. Introduction

${CO}_{2}$ corrosion, also known as “sweet corrosion”, is a type of corrosion caused by carbon dioxide dissolving to form carbonic acid. When ${CO}_{2}$ dissolves in water, the resulting carbonic acid dissociates to produce $H^{+}$ and ${HCO}_{3}^{-}$ ions, which play a key role in the corrosion process. Carbonic acid is highly corrosive to metallic materials, particularly steel. Under the same pH conditions, the carbonic acid in a ${CO}_{2}$-containing solutions is often more corrosive than hydrochloric acid. ${CO}_{2}$ corrosion significantly reduces the lifespan of gathering pipelines and equipment, with corrosion rates exceeding 7 mm/year [1].

${CO}_{2}$ corrosion encompasses two parts: uniform corrosion and localized corrosion [2], each with different mechanisms. The principles of uniform corrosion include anodic and cathodic corrosion. Anodic corrosion primarily involves the electrochemical oxidation of iron. Specifically, iron undergoes several oxidation steps at the anode. First, iron reacts with hydroxide ions to form iron hydroxide, releasing an electron. Subsequently, iron hydroxide further reacts with hydrogen ions to produce ferrous ions and water, also releasing an electron. Additionally, iron hydroxide can react with hydroxide ions to generate ferrous ions and water while releasing an electron. These reactions collectively drive the oxidation of iron, leading to the formation of ferrous ions and accelerating the corrosion process. The cathodic corrosion process involves two main reduction reactions: non-catalytic hydrogen ion reduction and catalytic reduction of surface-adsorbed hydrogen ions. In the non-catalytic hydrogen ion reduction reaction, when the pH is below 4, hydrogen ions are reduced at the cathode to form hydrogen gas and water. When the pH is between 4 and 6, carbonic acid is reduced to hydrogen ions and bicarbonate ions. When the pH is greater than 6, bicarbonate ions are reduced to hydrogen gas and carbon dioxide. On the other hand, in the catalytic reduction of surface-adsorbed hydrogen ions, carbon dioxide first adsorbs onto the metal surface and reacts with water to form carbonic acid. Carbonic acid then decomposes during the cathodic reduction process to produce hydrogen ions and bicarbonate ions, with hydrogen ions ultimately being converted to hydrogen gas and water. These reactions illustrate the primary cathodic reaction mechanism of the ${CO}_{2}$ corrosion process [3]. The overall corrosion reaction involves iron reacting with carbon dioxide and water to produce ferrous carbonate and hydrogen gas. Localized corrosion is the main cause of ${CO}_{2}$ corrosion [4]. When ${CO}_{2}$ acts as the corrosive medium, corrosion products such as Fe${CO}_{3}$ form a film on the steel surface. Due to uneven coverage, this can create galvanic or occluded cells, thereby accelerating localized corrosion. The phenomenon of localized corrosion includes pitting corrosion [5], crevice corrosion [6], and flow-induced corrosion [7], all of which result from galvanic coupling.

The typical models for predicting ${CO}_{2}$ corrosion include empirical models [8], semi-empirical models [9], and mechanistic models [10]. Empirical models are entirely based on laboratory data and field data. The most famous one is Norway’s Norsok M506 model [11], which is an empirical model established based on low-temperature experimental data and high-temperature field data. This model has become an important standard for the Norwegian petroleum industry for selecting materials to resist ${CO}_{2}$ corrosion and determining the corrosion allowance design. Semi-empirical models are currently the most widely used type of prediction model, mainly including SHELL 75 [12], SHELL 91 [13], SHELL 93 [14], and SHELL 95 [15]. Among them, the initial SHELL 75 only considered the effects of temperature and ${CO}_{2}$ partial pressure; SHELL 91 added considerations for pH, ${Fe}^{2+}$ concentration, corrosion product films, and crude oil; SHELL 93 modified the correction factors of the SHELL 91 model and preliminarily proposed the influence of flow rate; the SHELL 95 model includes the corrosion reaction kinetics process that is independent of flow rate and the mass transfer process that is related to flow rate. At temperatures below 80 °C, the model’s predictions align well with the loop experiment results, and different influencing factors are used for different steel types based on their structure and composition.

Mechanistic models are mainly based on the uncoupled description of electrochemical, chemical, and mass transfer processes in corrosion and have been well-validated in cyclic voltammetry. Because the mechanistic model starts from basic physical principles and uses classical dynamic formulas to calculate corrosion rates, it is relatively easier to correct the defects in existing prediction models. In 1996, Nesic developed a mechanistic model based on the reaction kinetics of the ${CO}_{2}$ corrosion process, studying the transfer processes of ions and electrons in the product film. Therefore, this model can not only predict corrosion rates, but also ion concentrations and flow rates. The prediction results lie between those of the De Waard model and the Dugstad model. However, this model does not account for the influence of the medium transfer process; therefore, it is only applicable in situations without corrosion product films. In low temperature and low pH conditions, the effect of FeC on corrosion rates needs to be considered [16].

Traditional empirical models, semi-empirical models, and mechanistic models have limitations in handling complex nonlinear relationships, data processing capabilities, adaptability, reliance on physical mechanisms, predictive performance, and comprehensively considering multiple influencing factors. In contrast, machine learning algorithms offer strong nonlinear modeling capabilities, large-scale data processing, adaptability, the advantage of not requiring explicit physical mechanisms, superior predictive performance, and the ability to comprehensively consider multiple influencing factors. These advantages effectively address the shortcomings of traditional models, significantly enhancing the accuracy and applicability of ${CO}_{2}$ corrosion prediction.

Abbas [17] et al. conducted neural network modeling for pipeline steel under high-pressure conditions. They developed and utilized various Matlab functions for transfer and training in their neural network model. Results demonstrated that the tansig transfer function outperformed the logsig transfer function in predicting high-pressure ${CO}_{2}$ corrosion. The model’s prediction results fell within the 95% confidence interval, demonstrating high predictive accuracy. Ossia [18] employed feed-forward neural networks, gradient boosting machines, and random forests to estimate the corrosion defect depth growth in aging pipelines. The research results showed that after applying PCA transformation to these algorithms, their model accuracy was three to five times higher than those without PCA transformation, with the PCA-GBM showing the best performance. Yan et al. [19] introduced a machine learning-based method for modeling the corrosion characteristics of low-alloy steel in oceanic environments. They evaluated the correlation among material types, environmental conditions, and corrosion rate, selected dominant factors as input variables and constructed an optimized random forest model. The model attained determination coefficients of 0.94 during training and 0.73 during testing. Aghaaminiha et al. [20] employed machine learning modeling to study the variation in carbon steel corrosion rates over time under the influence of corrosion inhibitors. They used Artificial Neural Networks, Random Forests, Support Vector Machines, and K-Nearest Neighbors for modeling. The results indicate that the severity of corrosion environments varies with factors such as temperature, ${CO}_{2}$ partial pressure, salinity, and ion intensity. The RF model outperformed other models by accurately forecasting the temporal patterns of carbon steel corrosion, achieving a mean squared error between 0.005 and 0.096.

Although the aforementioned machine learning methods perform well in fitting common ion concentrations, temperature, and pressure, they still have limitations, such as insufficient adaptability to complex and variable datasets and failure to fully consider complex interactions between multiple factors. This study addresses significantly diverse datasets, including scenarios with the simultaneous presence of acidic gases, water, and different materials, employing machine learning models like RF and XGBoost to separately model the corrosion rate and severity. This approach helps enhance the model’s personalized, predictive accuracy for corrosion behavior across different materials, optimize resource utilization, gain deeper insights into variations in corrosion response among different categories, and provide more accurate predictions and explanations for effective corrosion control.

The general structure of the paper is outlined as follows: Section 2 introduces the machine learning algorithms used and discusses the evaluation metrics of the models. Section 3 summarizes the data and analyzes the correlations between various corrosion features using Pearson correlation coefficients. Section 4 presents the results of the model fitting and evaluates the model’s performance. Finally, Section 5 encapsulates the principal discoveries of the research.

## 2. Overview of Algorithms and Evaluation Metrics for Models

Machine learning techniques are categorized into supervised and unsupervised learning methods. Supervised learning involves training a model with known data and their corresponding labels and mapping the input data to the label. Supervised learning includes classification problems and regression problems. In classification, the results correspond to discrete values, such as using X-rays in medical imaging to detect tumors and classify images as either having or not having tumors. Regression, on the other hand, deals with continuous values, like predicting future stock prices using historical stock price data and related features. Unsupervised learning, on the other hand, involves learning and discovering patterns in the training samples without knowing the corresponding labels. It can be used to group customers with similar shopping habits into different segments, thus facilitating better marketing strategy development. When predicting corrosion rates using machine learning methods, supervised learning algorithms are commonly employed.

In this study, common machine learning algorithms such as SVM, GBDT, and RF were applied to the organized data. The data were randomly sampled, with 80% used as the training set to establish a well-performing model and the remaining 20% used as the test set to evaluate the model’s generalization performance.

### 2.1. Introduction to Machine Learning Algorithms

#### 2.1.1. Overview of the Random Forest Algorithm

The Random Forest [21] algorithm is a typical representative of ensemble learning algorithms. Based on statistical theory, it employs bootstrap sampling to extract multiple sample sets from the training data and uses these sample sets to construct individual decision tree models. These decision trees are then aggregated together, and the final result is obtained through majority voting or averaging. Decision tree algorithms are top-down tree structure classifiers that generate division rules through logical induction of the training dataset. The main forms of decision trees include binary trees and multiway trees. The combination of decision tree algorithms and ensemble learning concepts forms the foundation of the Random Forest algorithm.

The key to the decision tree algorithm lies in the tree construction process, which includes tree generation and pruning. After generating a decision tree, feature selection is the most critical part of the tree division process. Different feature evaluation methods select features with varying abilities to divide samples, significantly impacting the final classification performance of the decision tree. Common techniques for evaluating features include information gain, information gain ratio, and the Gini index.

The concept of information gain is derived from information entropy, a concept introduced by American mathematician Shannon in his 1948 information theory, which is used to measure the degree of order and disorder in a system. Let D be a dataset, and A be a feature in dataset D. Feature A has k distinct values {$a_{1}$, $a_{2}$, …, $a_{k}$}. Feature A can divide D into k subsets {$D_{1}$, $D_{2}$, …, $D_{k}$}. The information gain of feature A is defined as:(1)$$Gain(D,A)=E(D)-E_{A}(D)$$

The information gain ratio of feature A represents the ratio of the information gain of feature A to its intrinsic information in dataset D. The calculation formula is as follows:(2)$$Gain_{R}(D,A)=Gain(D/A)/IntI(A)$$

The Gini index is a measure of data purity. A higher Gini index indicates greater data uncertainty, whereas a lower Gini index suggests higher data purity. Under the condition A = $a_{j}$ (where j = 1, 2, …, k), dividing dataset D into two parts $D_{1}$ and $D_{2}$, the Gini index is calculated as:(3)$$Gini(D,A=a_{j})=(|D_{1}|/n)Gini(D_{1})+(|D_{2}|/n)Gini(D_{2})$$

The decision tree algorithm does not require prior knowledge, and the model is simple and can quickly extract rules from the data. However, as a standalone classifier, decision trees perform poorly and have low generalization ability. In 1979, Dasarathy [22] introduced the concept of ensemble classifiers, marking the origin of ensemble learning. Ensemble learning can be understood simply as addressing complex classification problems by combining multiple diverse individual classifiers using specific methods and integrating predictions from all classifiers to make final decisions. Appendix A provides the detailed code for the Random Forest algorithm used in this study.

#### 2.1.2. Introduction to XGBoost Algorithm Principle

XGBoost (Extreme Gradient Boosting) is a boosting algorithm based on regression trees, proposed by Dr. Tianqi Chen in 2016 [23]. As an open-source framework, XGBoost improves upon the boosting algorithm in GBDT (Gradient Boosting Decision Tree). While GBDT optimizes by learning the gradient of the loss function, XGBoost learns the difference in the second-order Taylor expansion of the loss function. Additionally, XGBoost includes a regularization term in the cost function to control the model complexity, ensuring extremely high speed while maintaining high accuracy. The core idea of XGBoost is to continuously add trees for feature splitting, where each added tree essentially learns a new function to fit the residuals of the previous prediction. Once the training is completed and k trees are obtained, the score of a sample is predicted based on the feature-specific leaf nodes in each tree. Finally, by summing the scores from all trees, the prediction for the sample is obtained.

The goal of XGBoost is defined by combining two components, the loss function, and regularization term, aimed at balancing the model’s fitting capability and complexity. The calculation formula for this objective function is as follows:(4)$$Obj=\sum_{i=1}^{n}l(y_{i},\hat{y_{i}})+\sum_{j=1}^{t}\Omega (f_{j})$$

The regularization part sums up the complexities of all t trees and serves as the regularization term in the objective function. The predicted value of the model for the i-th sample in the t-th tree is as shown in Equation (5):(5)$${\hat{y_{i}}}^{t}=\sum_{k=1}^{t}f_{k}(x_{i})={\hat{y_{i}}}^{\left(t-1\right)}+f_{t}(x_{i})$$

In the above formula: $y_{i}$ is the predicted result of sample i after the t-th update; $f_{t}(x_{i})$ is the model prediction result of the t-th tree; ${\hat{y_{i}}}^{\left(t-1\right)}$ is the prediction residual of sample i before the t-th update; $l(y_{i},\hat{y_{i}})$ is the loss function, where $\hat{y_{i}}$ is the predicted value of the entire model for the i-th sample, and y is the true value of the i-th sample.

The complexity of the tree can be denoted by $\Omega (f_{t})$, aiming to constrain the tree’s structure to prevent overfitting.
(6)$$\Omega (f_{t})=\gamma T+\frac{1}{2}\lambda \sum_{j=1}^{T}{\omega_{j}}^{2}$$
where γ is a tuning hyperparameter, ${\omega_{j}}^{2}$ denotes the squared 12 norm of the leaf node weights ω on the tree.

The XGBoost algorithm is a potent tool extensively utilized in the realm of artificial intelligence. It not only inherits the advantages of Boosting algorithms but also controls model complexity through regularization to prevent overfitting, introducing regularization terms in the cost function. Its unique parallel processing structure preprocesses the dataset by sorting and handling it in block structures, thus achieving efficient feature selection. Under the condition that the objective function is twice differentiable, XGBoost supports user-defined objective functions, demonstrating flexibility and scalability. Additionally, its built-in cross-validation functionality helps determine the optimal number of iterations, further optimizing the model performance. Appendix A provides the detailed code for the XGBoost algorithm used in this study.

### 2.2. Evaluation of Multiple Model

#### 2.2.1. Performance Evaluation Metrics for Regression Models

To assess the performance of predictive models built using various algorithms, this paper employs the coefficient of determination ($R^{2}$) and mean squared error (MSE) as evaluation metrics. The coefficient of determination, $R^{2}$, is a statistical measure used to evaluate the goodness of fit of regression models, quantifying the proportion of the variance in the target variable that is explained by the independent variables. The calculation formula is as follows:(7)$$R^{2}=1-\frac{\sum_{i}(y_{i}-f_{i}{)}^{2}}{\sum_{i}(y_{i}-\bar{y}{)}^{2}}$$
where $y_{i}$ is the actual value of the i-th sample, $f_{i}$ is the predicted value of the ith sample, and $\bar{y}$ is the mean value of the actual values. The coefficient of determination typically ranges between 0 and 1, with larger values indicating better predictive capability of the model.

The mean squared error measures the average of the squared differences between the model’s predicted values and the actual values. The expression is:(8)$$MSE=\frac{1}{n}(y_{i}-{\hat{y}}_{i}{)}^{2}$$
where n is the number of samples, $y_{i}$ is the actual value of the ith sample, and ${\hat{y}}_{i}$ is the model’s predicted value for the ith sample. A lower MSE value signifies a closer alignment between the model’s predictions and the actual values, indicating superior model accuracy.

#### 2.2.2. Evaluation Metrics for Classification Performance

In binary classification tasks, the confusion matrix is considered one of the most practical evaluation tools. In such tasks, the model’s predicted results and actual results are represented by 0 and 1, respectively, with N and P used to denote these two numbers. T and F represent correct and incorrect predictions, respectively. The confusion matrix includes the following sample quantity information: True Positive (TP) represents samples correctly classified as belonging to the positive class, False Positive (FP) represents samples incorrectly classified as belonging to the positive class, False Negative (FN) represents samples incorrectly classified as not belonging to the positive class, and True Negative (TN) represents samples correctly classified as not belonging to the positive class. Multiclass problems can be understood as a collection of multiple binary classification problems. In multiclass problems, four evaluation metrics are commonly used to assess model performance: accuracy, precision, recall, and F1-score.

Precision refers to the proportion of samples predicted as positive by the algorithm that are positive. A high precision indicates that the model accurately predicted most of the positive samples, with a few negative samples mistakenly classified as positive. Its mathematical formula is expressed as
(9)$$Precision=\frac{TP}{TP+FP}$$

Recall represents the proportion of actual positive samples in the dataset that are correctly identified as positive by the model. The formula for recall is expressed as
(10)$$Recall=\frac{TP}{TP+FN}$$

The F1-score comprehensively considers the classifier’s accuracy and coverage in the positive class, particularly suitable for scenarios requiring a balance between precision and recall.
(11)$$F1-Score=\frac{2}{\frac{1}{Precision}+\frac{1}{Recall}}$$

The F1-score includes three variants: macro-F1, micro-F1, and weighted-F1. Macro-F1 calculates the Precision, Recall, and F1-score for each class and then takes the average with equal weights, treating each class equally, regardless of class imbalance. Micro-F1 adds up the TP, FP, and FN for each class, applies the F1-score formula for binary classification, and combines the performance of all classes into a single overall indicator, ignoring differences in sample quantities for each class. Weighted-F1 assigns different weights to each class based on the proportion of samples in each class, calculates the weighted average of the F1-scores for each class, considers class imbalance, and pays more attention to classes with larger sample sizes.

Accuracy refers to the proportion of correctly predicted instances by the algorithm model in the overall dataset, where the predicted class matches the actual class (0 for 0, 1 for 1). The formula is as follows:(12)$$Accuracy=\frac{TP+TN}{TP+TN+FP+FN}$$

This paper adopts accuracy and F1-score as the evaluation metrics for the classification model. These metrics not only comprehensively assess the model’s classification accuracy but also strike a balance between precision and recall, providing a holistic perspective for evaluating model performance.

## 3. Data Mining

### 3.1. Data Preprocessing

Before establishing machine learning models, it is crucial to meticulously organize and analyze data using data analysis techniques to deeply explore the broad characteristics and profound implications within the data, revealing underlying patterns essential for constructing subsequent models. This compilation includes 41 articles discussing the corrosion behavior of API 5L X65 carbon steel under supercritical and liquid carbon dioxide phases in the presence of water and sulfur dioxide [24] and the effect of exposure time on the corrosion rates of X70 steel in supercritical $CO_{2}$/$SO_{2}$/$O_{2}$/$H_{2}O$ environments [25]. The dataset comprises 248 sample points, each containing 10 features: material, chromium content, water content, oxygen content, sulfur dioxide content, nitrogen dioxide content, hydrogen sulfide content, pressure, temperature, and time. Regression results indicate corrosion rates, while classification results indicate corrosion severity. Detailed data summaries are presented in Table 1, with gas units measured in ppmv.

Table 1 provides a statistical analysis of the influencing factors in the data. Since the material does not involve statistical items such as maximum and minimum values, it was not included in the analysis. The table shows that the data are relatively dispersed. According to the analysis of the key factors in the overall dataset, such as corrosion rate and severity level, the average corrosion rate is 0.50 mm/a, with a minimum of 0.00 mm/a and a maximum of 26.00 mm/a. This indicates that the maximum corrosion rate is an outlier, occurring in only a few cases. Therefore, the analysis mainly focuses on data within the 0–6 mm/a range. The corrosion severity level ranges from 0 to 3, with an average of 2.14 and a standard deviation of 1.17, indicating a relatively dispersed distribution of severity levels. These statistical insights help to better understand the overall corrosion situation, identify trends and anomalies in the data, and provide a reference for formulating effective corrosion control measures.

### 3.2. Correlation Analysis

Data mining requires not only the analysis of individual factors but also the study of the correlation between influencing factors and the impact of various features on the corrosion rate. The renowned statistician Karl Pearson introduced the Pearson correlation coefficient in the 1880s [26], which is used to measure the correlation between two variables, X and Y. The expression is as follows:(13)$$\gamma (X,Y)=\frac{Cov(X,Y)}{\sqrt{Var\left[x\right]Var[Y]}}$$

In the equation, Cov(X,Y) signifies the covariance between X and Y, Var(X) indicates the variance of X, and Var(Y) indicates the variance of Y.

The calculated correlation coefficient ranges from −1 to 1. If the correlation coefficient between two corrosion-influencing factors is 0, it means that the two variables are relatively independent and uncorrelated. When the correlation coefficient between the two corrosion parameters is between 0 and 1, it indicates a positive correlation. When the correlation coefficient is between 0 and −1, it indicates a negative correlation. If the absolute value of the correlation coefficient is higher, the relationship between the corrosion parameters will also be stronger.

According to this theory, the correlation coefficients between various corrosion factors were calculated, generating the correlation coefficient matrix heatmap shown below:

Based on the data presented in Figure 1, the correlation coefficients among the corrosion factors are low, suggesting that these factors operate relatively independently. As a result, feature reduction is deemed unnecessary. Acidic gases, such as sulfur dioxide ($SO_{2}$), nitrogen dioxide ($NO_{2}$), and hydrogen sulfide ($H_{2}S$), significantly impact the corrosion rate and severity of metal materials, thereby accelerating the corrosion process. Consequently, when developing corrosion protection strategies, it is essential to consider the presence of these gases and their potential effects on metal materials.

Figure 2 further illustrates the relationship between the influencing factors and both the corrosion rate and severity. As depicted in Figure 2a, water ($H_{2}O$) and acidic gases like $SO_{2}$ and $H_{2}S$ exert a pronounced effect on the corrosion rate of steel pipes, surpassing the impacts of temperature and pressure. These acidic gases are particularly corrosive, significantly accelerating the corrosion process and, consequently, the corrosion rate. In contrast, Figure 2b reveals that although acidic gases continue to substantially influence the corrosion severity, the effect of water is relatively minor. This disparity could be attributed to the direct and intense impact of acidic gases on the corrosion rate, whereas corrosion severity considers the extent and distribution of corrosion. The influence of water may be mitigated by other factors, thereby diminishing its relative effect on corrosion severity.

## 4. Discussion

### 4.1. ML Models for Carbon Dioxide Corrosion Rate

In this study, widely used algorithms in relevant fields, such as RF, KNN, SVM, and XGBoost, were employed to predict the carbon dioxide corrosion rate. To comprehensively assess the performance of these six algorithms, MSE was selected as the primary evaluation metric. Figure 3 displays the fitting performance of the different algorithms in predicting the corrosion rates. Through these charts, we can delve into the similarities and differences in the performance of each algorithm.

Figure 3 illustrates the corrosion rate fitting graphs for various machine learning algorithms, where the left axis represents the actual corrosion rate and the right axis represents the algorithm-predicted corrosion rate. The X = Y line in the graph serves as a reference, representing an ideal scenario in which the predicted corrosion rate matches the actual corrosion rate. Ideally, all prediction points should cluster around this line, indicating a perfect alignment between the model predictions and actual observations. The distribution of the corrosion rates predicted by the machine learning algorithms on both the training and test sets around the X = Y line suggests reasonable proximity to the actual values, indicating a degree of consistency. However, for a more thorough evaluation, additional metrics and domain-specific knowledge should be considered.

A comparative analysis reveals that the fitting performance of SVM in Figure 3a and LightGBM in Figure 3b is the worst in the 0–6 mm/a corrosion rate range. In Figure 3c–e, the GBDT, XGBoost, and KNN models perform well in the training set within the 0–6 mm/a corrosion rate range, with the predicted values closely aligned with the actual values. However, the prediction performance of these models in the test set is not ideal, significantly deviating from the X = Y line. The RF model shows relatively good fitting performance in both the training and test sets, making it the best-fitting model among these algorithms.

To comprehensively assess the performance of these models, we employ the coefficient of determination and mean squared error as evaluation metrics to calculate the performance indicators for six different prediction models.

Upon analyzing Figure 4, it was observed that the $R^{2}$ values for the training set are generally higher than those for the test set. Both SVM and LightGBM exhibit poor performance across both sets, with $R^{2}$ values below 0.6 in the training set. In contrast, KNN, GBDT, and XGBoost demonstrate strong performance on the training set but exhibit lower $R^{2}$ values on the test set, all falling below 0.8, suggesting a certain degree of overfitting. The RF model, however, achieves the highest $R^{2}$ value on the test set, indicating more consistent performance across different datasets and superior generalization capability.

MSE is a metric that quantifies the difference between the predicted and actual values by calculating the average of the squared differences between them. A lower MSE signifies smaller discrepancies between the model’s predictions and actual outcomes, reflecting better model performance. The analysis Figure 5 reveals that SVM consistently yields higher MSE values on both the training and test sets, indicating substantial prediction errors in estimating corrosion rates, making it the weakest performer among the six models. In contrast, while GBDT, XGBoost, and KNN show lower MSE values on the training set, their MSE values increase significantly on the test set, suggesting overfitting. The RF model, on the other hand, exhibits the lowest MSE value on the test set.

In summary, the RF model outperforms the others, with the highest $R^{2}$ value and the lowest MSE, indicating its superior ability to predict corrosion rates. Comparatively, the GBDT and XGBoost models offer slightly weaker predictive performance, while the SVM model proves to be the least effective.

### 4.2. ML Models for Carbon Dioxide Corrosion Severity

The comparison of the prediction results of four different models: Random Forest (RF), K-Nearest Neighbors algorithm (KNN), Support Vector Machine (SVM), and LightGBM, as illustrated in Figure 6.

From Figure 6, KNN exhibits the lowest average accuracy of 77.65%, followed by SVM, with a higher average accuracy of 93%. RF, however, outperforms both, achieving the highest average accuracy of 98.2%. Furthermore, RF also secures the highest average F1-score for classifying corrosion severity, with a score of 0.98.

Based on the results analysis from Table 2 and Figure 7, the following conclusions can be drawn: among the four common machine learning algorithms used for $CO_{2}$ corrosion severity prediction, RF algorithm performs the best, exhibiting superior predictive performance with only slight errors observed in corrosion severity 2. Conversely, the predictive performance of KNN algorithm is poor, particularly with significant errors observed at ${CO}_{2}$ corrosion severities of 1 and 2. The SVM algorithm performs better compared to KNN, with significant improvement in prediction accuracy at corrosion severities of 1, 2, and 3. While LightGBM excels in predicting the corrosion severity of 0 and 1, it shows slight weaknesses in grades 2 and 3. The accuracy and F1-scores of the RF, SVM, and LightGBM models all exceed 90% and 0.9, while for the KNN model, they are only 82% and 0.81, respectively, indicating its poor performance in discriminating corrosion severities 1 and 2. This suggests that the RF model outperforms SVM, KNN, and LightGBM machine learning models in predicting $CO_{2}$ corrosion severity.

## 5. Conclusions

Carbon dioxide corrosion poses a significant threat to equipment safety and longevity. Accurate prediction of corrosion rates is essential for effective material selection and equipment maintenance. Traditional corrosion prediction models often fall short in accounting for the complex interactions among multiple factors, resulting in limited accuracy, adaptability, and generalization. To overcome these limitations, this study leverages machine learning techniques to process organized data and performs a detailed analysis of the factors influencing corrosion through correlation analysis. The key findings are as follows:

Spearman correlation analysis reveals that, within this dataset, water is the most influential factor affecting the corrosion rate. Additionally, acidic gases such as hydrogen sulfide, sulfur dioxide, and other sulfides exhibit a significant positive correlation with the corrosion rate. Conversely, the temperature, pressure, and material show a negative correlation with the corrosion rate, although their impact is relatively minor. Acidic gases significantly affect both the rate and severity of corrosion.

In the model validation and evaluation phase, fitting carbon dioxide corrosion rates with six different algorithms yielded the following results: Support Vector Machine (SVM) showed the poorest performance, with suboptimal results on both the training and test sets. In contrast, the XGBoost, Gradient Boosting Decision Tree (GBDT), and K-Nearest Neighbors (KNN) algorithms performed well on the training set but exhibited weaker performance on the test set. The Random Forest (RF) algorithm demonstrated superior overall performance, excelling in both training and test set evaluations.

For the classification of carbon dioxide corrosion levels, the F1-score and accuracy were utilized as evaluation metrics. The results indicate that the KNN algorithm performed the least effectively, with an accuracy of 82% and an F1-score of 0.81. In contrast, the RF algorithm achieved the highest performance for both metrics, with an accuracy of 99% and an F1-score of 0.99.

Models constructed using machine learning can predict corrosion rates and severity, thereby optimizing material selection, reducing equipment maintenance costs, and extending equipment lifespan. Analyzing machine learning models contributes to a deeper understanding of corrosion mechanisms in different materials, laying the groundwork for further research.

## Figures and Tables

**Figure 1 materials-17-04046-f001:**
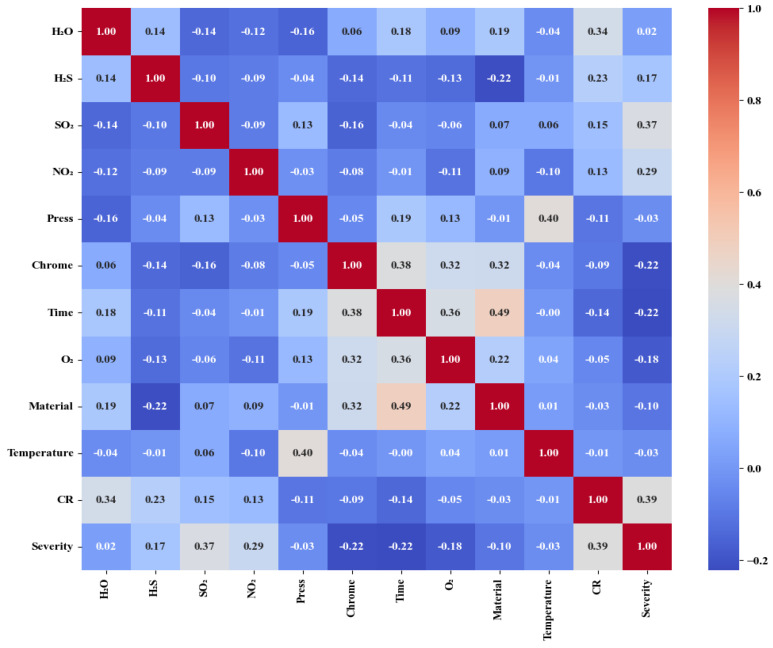
Correlation matric heat map.

**Figure 2 materials-17-04046-f002:**
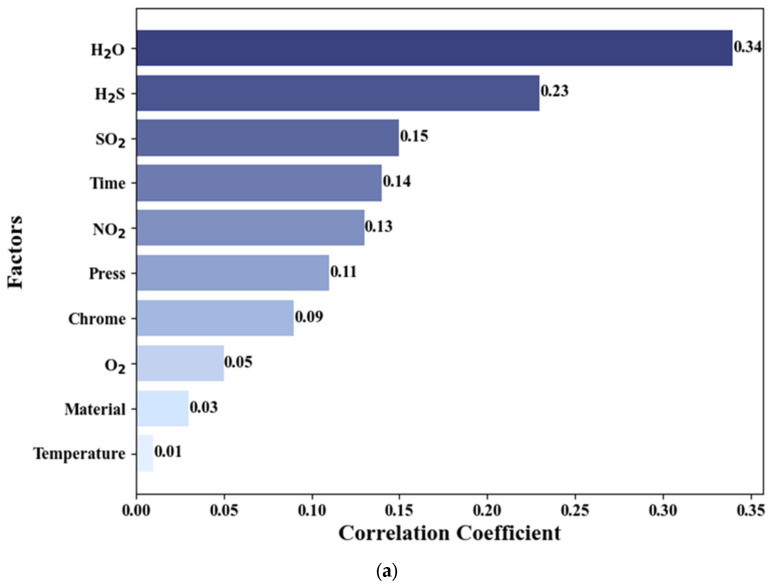
Bar chart of correlation for target factors: (**a**) corrosion rate and (**b**) corrosion severity.

**Figure 3 materials-17-04046-f003:**
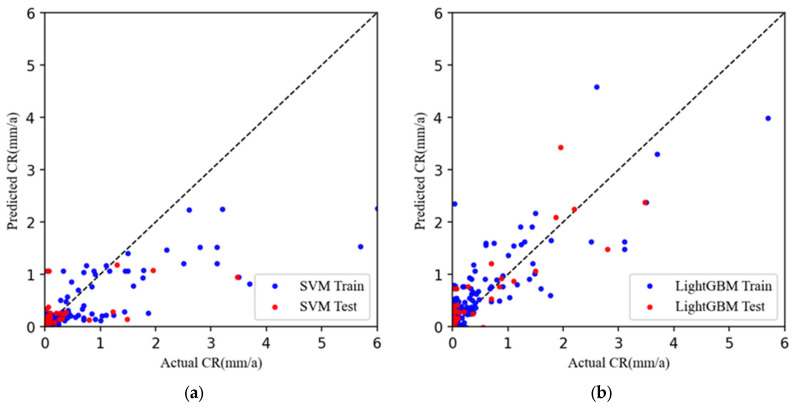
Fit graphs of corrosion rates with different machine learning algorithms (**a**) SVM, (**b**) LightGBM, (**c**) GBDT, (**d**) XGBoost, (**e**) KNN, (**f**) RF. Dashed line represents the agreement between actual and predicted corrosion rates.

**Figure 4 materials-17-04046-f004:**
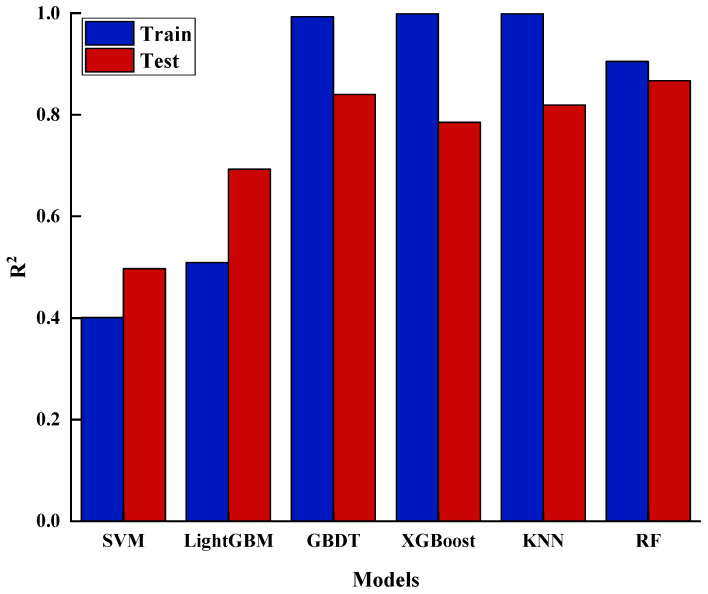
$R^{2}$ value charts for different models.

**Figure 5 materials-17-04046-f005:**
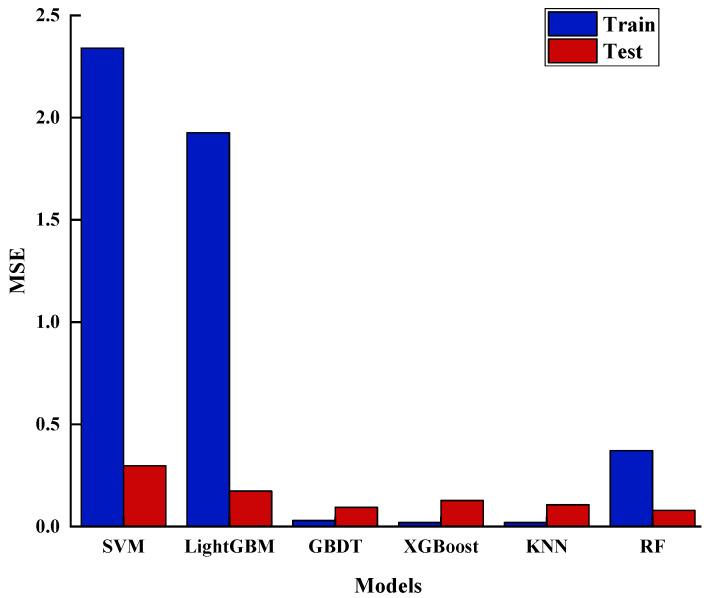
MSE charts for different models.

**Figure 6 materials-17-04046-f006:**
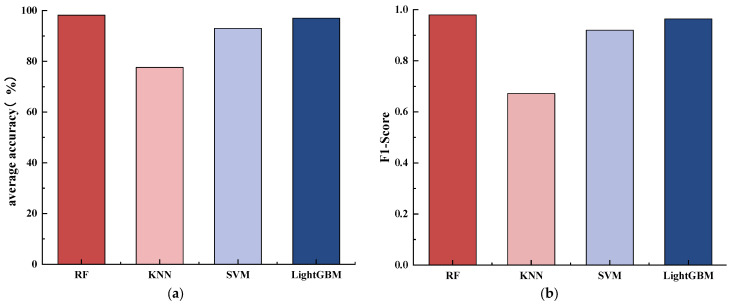
Comparison of corrosion severity level classification results among different models: (**a**) accuracy and (**b**) F1-Score.

**Figure 7 materials-17-04046-f007:**
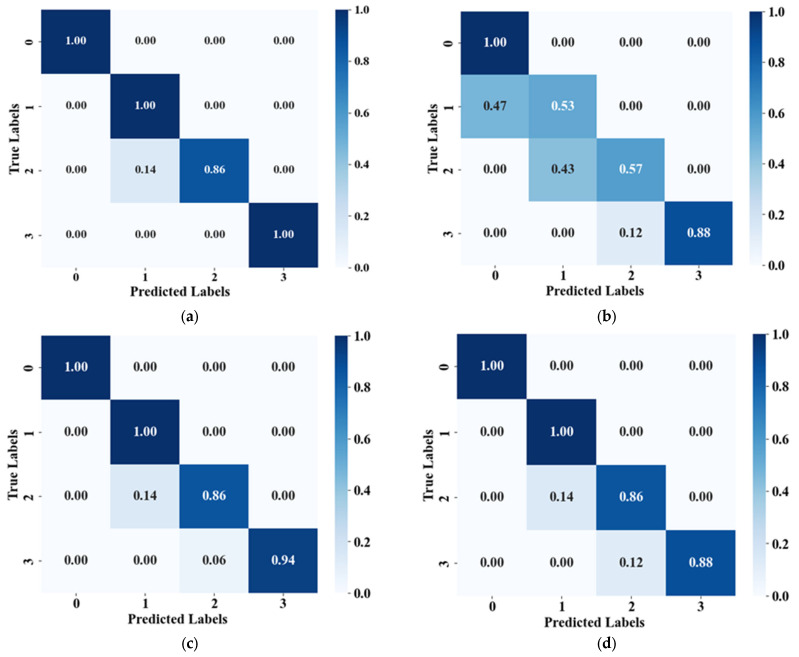
The confusion matrix comparison chart for predicting ${CO}_{2}$ corrosion severity using different algorithms (**a**) RF, (**b**) KNN, (**c**) SVM, and (**d**) LightGBM.

**Table 1 materials-17-04046-t001:** Data Summary.

	Factor	Chrome	Temperature	H_2_O	O_2_	SO_2_	NO_2_	Time	CR	Severity
Statistics										
Mean		0.09	48.27	29.50	5208.75	2034.76	34.30	191.01	0.50	1.14
Std		0.13	20.37	62.88	11,634.29	5961.86	127.01	254.14	1.81	1.17
Min		0.00	25.00	0.08	0.00	0.00	0.00	1.50	0.00	0.00
25%		0.01	40.00	0.96	0.00	0.02	0.00	48.00	0.03	0.00
50%		0.04	50.00	2.73	20.00	0.08	0.00	120.00	0.08	1.00
75%		0.11	50.00	34.00	1000.00	0.33	0.00	168.00	0.30	2.00
Max		0.54	200.00	400.00	47,000.00	26.00	1000.00	1512.00	26.00	3.00

**Table 2 materials-17-04046-t002:** The comparison of evaluation results of corrosion severity among different models.

Algorithm	Accuracy (%)	F1-Score
RF	99	0.99
KNN	82	0.81
SVM	92	0.92
LightGBM	96	0.96

## Data Availability

Due to privacy reasons, the data provided in this study cannot be publicly disclosed. If needed, please contact the corresponding author for more information.

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
