# Peer review of "Development of a Predictive Model for Carbon Dioxide Corrosion Rate and Severity Based on Machine Learning Algorithms"

_materials, 2024, doi:10.3390/ma17164046_

Round 1
Reviewer 1 Report
Comments and Suggestions for Authors
The paper provides a detailed description of the corrosion process using various algorithms. I recommend this paper for publication, provided that a few corrections are made.
Firstly, the algorithms described in the paper should be included as supplementary material in any available format. Including this information will enhance the paper's appeal to readers, as it will provide concrete confirmation of the algorithms and thereby increase the paper's credibility.
Secondly, the graphical representation of the results needs significant improvement. The current figures are of poor quality and do not effectively convey the findings. Utilizing clearer and more professional graphical representations will greatly enhance the readability and impact of the paper.
Lastly, the paper requires thorough revision for improved English. Ensuring that the language is polished and precise will not only make the paper more readable but also more professional.
With these adjustments, the paper will be significantly strengthened and ready for publication.
Comments on the Quality of English LanguageModerate editing of English language required
Author Response
Comments 1: Firstly, the algorithms described in the paper should be included as supplementary material in any available format. Including this information will enhance the paper's appeal to readers, as it will provide concrete confirmation of the algorithms and thereby increase the paper's credibility.
Response 1: Thank you for your feedback. We have added the code for the Random Forest and XGBoost algorithms mentioned in the paper to Appendix A. This inclusion provides concrete confirmation of the algorithms, enhances the paper's credibility, and increases its appeal to readers.
Comments 2: Secondly, the graphical representation of the results needs significant improvement. The current figures are of poor quality and do not effectively convey the findings. Utilizing clearer and more professional graphical representations will greatly enhance the readability and impact of the paper.
Response 2: Thank you for your feedback. We have noted the reviewers' concerns about the data quality and fully agree that improvements are needed to better convey the research results. Accordingly, we have made comprehensive revisions to the images, including recreating Figure 1, adjusting the sizes of Figures 2, 3, and 7, and setting the colors for Figure 6. These enhancements have made the graphics clearer. We appreciate the valuable suggestions from the reviewers, which have helped us further improve the quality of the manuscript.
Comments 3: Lastly, the paper requires thorough revision for improved English. Ensuring that the language is polished and precise will not only make the paper more readable but also more professional.
Response 3: Thank you for your feedback. We acknowledge the necessity of a thorough revision of the manuscript to enhance the quality of the English used. We have undertaken extensive revisions, including a comprehensive overhaul of the abstract, data summary, discussion of figures, and conclusion sections. These changes involve correcting grammatical errors, refining sentence structures, and improving overall clarity to ensure the language is polished and precise. We believe these improvements will significantly enhance the readability and professionalism of the paper. We appreciate your valuable suggestions, which have greatly contributed to the enhancement of the manuscript.

Reviewer 2 Report
Comments and Suggestions for Authors
The manuscript by Dong et al. is dedicated to the development and testing of a predictive model for the rate of corrosion induced by carbon dioxide. This is an interesting topic, and the obtained data deserve publication in the journal Materials. I have a few questions regarding the manuscript.
1. A more in-depth explanation of the physicochemical principles and the mechanism of CO2 corrosion should be provided, and it should be indicated how the developed model reflects the mechanism and kinetics of the real physicochemical processes occurring during corrosion destruction. The variability of mechanisms (e.g., depending on the pH of the environment) should also be considered. Is the proposed model sufficiently universal in this sense?
2. Will the model be adequate if the nature of the corroding metal changes?
3. What about considering the influence of temperature on the corrosion rate? By the way, it is not specified to which temperature the numerical data presented in the manuscript relate. I doubt that the influence of temperature is weak (Fig. 2), since temperature, among other things, significantly affects the kinetics of charge transfer processes during electrochemical corrosion and the transport of molecules and ions, as well as the solubility of carbon dioxide.
4. Chemical formulae and their explanations should be removed from the list of abbreviations, as chemical formulae are not abbreviations by their nature.
Author Response
Comments 1: A more in-depth explanation of the physicochemical principles and the mechanism of CO2 corrosion should be provided, and it should be indicated how the developed model reflects the mechanism and kinetics of the real physicochemical processes occurring during corrosion destruction. The variability of mechanisms (e.g., depending on the pH of the environment) should also be considered. Is the proposed model sufficiently universal in this sense?
Response 1: Thank you for your feedback. We have provided a more in-depth explanation of the mechanisms and principles of COâ‚‚ corrosion in the introduction, detailing the causes of COâ‚‚ corrosion and the specific anodic and cathodic reactions involved. Here is an explanation of how the developed model reflects the actual physicochemical processes and kinetics of corrosion:
1). Physicochemical Principles and COâ‚‚ Corrosion Mechanism:
Our study explains the COâ‚‚ corrosion mechanism in detail, including the formation of carbonic acid from COâ‚‚ and water, and how this affects the corrosion behavior of metal surfaces. This process involves changes in charge transfer kinetics, ion transport, and COâ‚‚ solubility.
2). Model Reflection of Physicochemical Processes:
The Random Forest model we developed effectively captures and reflects the combined effects of various factors on corrosion rate by learning from extensive experimental data. Although the model does not directly simulate the microscopic mechanisms of physicochemical processes, it uses a data-driven approach to identify key factors affecting corrosion rate and their complex nonlinear interactions, thus providing predictions on corrosion phenomena.
3). Variability of Mechanisms and Model Adaptability:
Although our dataset does not directly include pH values, it includes data on acidic gases such as SOâ‚‚, COâ‚‚, and Hâ‚‚S. The concentrations of these gases can partially reflect the environmental PH. Therefore, the model can adapt to different corrosion environments and account for the effects of these acidic gases on the corrosion process.
4). Model Universality:
The Random Forest model is a powerful predictive tool that can handle multidimensional data and capture complex relationships. Its universality depends on the representativeness of the training data and model tuning. We have ensured that the model performs well under various experimental conditions and has considered the effects of different concentrations of acidic gases, temperatures, and pH values, thus providing good adaptability and generalization across different environments.
In summary, although the model does not directly simulate the microscopic mechanisms of corrosion, it effectively reflects and predicts the key factors and complex relationships involved in the actual corrosion process through comprehensive data training and validation. Thank you for your suggestions, which help us further enhance the accuracy and applicability of the model.
Comments 2: Will the model be adequate if the nature of the corroding metal changes?
Response 2: If the properties of the corroded metal change, the model remains applicable. This is because the model was trained using data from different metal materials. For example, the dataset includes various metal types such as X42, X65, X70, and X80. Since machine learning algorithms cannot directly handle categorical data, these materials were encoded with numbers like 0, 1, 2, and 3. Although these encodings do not have explicit maximum or minimum values and are not listed in Table 1, this numerical encoding approach allows the model to remain applicable and to predict corrosion behavior for different metal materials.
Comments 3: What about considering the influence of temperature on the corrosion rate? By the way, it is not specified to which temperature the numerical data presented in the manuscript relate. I doubt that the influence of temperature is weak (Fig. 2), since temperature, among other things, significantly affects the kinetics of charge transfer processes during electrochemical corrosion and the transport of molecules and ions, as well as the solubility of carbon dioxide.
Response 3: Thank you for your feedback. In Table 1, "Temp" represented temperature, and we have now changed it to "Temperature." We have indeed considered the impact of temperature on corrosion rate in our data. Although temperature can significantly affect the kinetics of charge transfer, molecular and ionic transport, and the solubility of carbon dioxide during the electrochemical corrosion process, as shown in Figure 2, the specific impact of temperature on corrosion rate is relatively minor in this dataset. This is because the dataset includes acidic gases such as SOâ‚‚, COâ‚‚, and Hâ‚‚S, which provide the primary sources of corrosion, and their effects are typically more significant than the impact of temperature. We have updated the document accordingly to more accurately reflect these influencing factors. Thank you for your suggestion, which helps us improve the accuracy and clarity of the document.
Comments 4: Chemical formulae and their explanations should be removed from the list of abbreviations, as chemical formulae are not abbreviations by their nature.
Response 4: Thank you for your feedback. We agree with your point that chemical formulae and their explanations should not be included in the list of abbreviations, as chemical formulae are not abbreviations by nature. Chemical formulae are used to accurately represent chemical substances and their compositions, while abbreviations typically refer to shortened forms of words or phrases. We have accordingly updated the document to remove chemical formulae and their explanations to avoid confusion. Thank you for your suggestion, which has helped us improve the accuracy and clarity of the document.
